# Investigation of the Correlation between Graves’ Ophthalmopathy and CTLA4 Gene Polymorphism

**DOI:** 10.3390/jcm8111842

**Published:** 2019-11-02

**Authors:** Ding-Ping Chen, Yen-Chang Chu, Ying-Hao Wen, Wei-Tzu Lin, Ai-Ling Hour, Wei-Ting Wang

**Affiliations:** 1Department of Laboratory Medicine, Chang Gung Memorial Hospital at Linkou, Taoyuan 33305, Taiwan; b9209011@cgmh.org.tw (Y.-H.W.); berry0908@cgmh.org.tw (W.-T.L.); s1223@adm.cgmh.org.tw (W.-T.W.); 2Graduate Institute of Biomedical Sciences, College of Medicine, Chang Gung University, Taoyuan 33302, Taiwan; 3Medical Biotechnology and Laboratory Science, Chang Gung University, Taoyuan 33302, Taiwan; 4Department of ophthalmology, Chang Gung Memorial Hospital at Linkou, Taoyuan 33305, Taiwan; yenchang@adm.cgmh.org.tw; 5Graduate Institute of Clinical Medical Sciences, College of Medicine, Chang Gung University, Taoyuan 33302, Taiwan; 6Department of Life Science, Fu Jen University, Taipei 24205, Taiwan; 022446@mail.fju.edu.tw

**Keywords:** Graves’ disease, Graves’ ophthalmopathy, CTLA4, gene polymorphism

## Abstract

Graves’ disease (GD) is an autoimmune inflammatory disease, and Graves’ ophthalmopathy (GO) occurs in 25–50% of patients with GD. Several susceptible genes were identified to be associated with GO in some genetic analysis studies, including the immune regulatory gene CTLA4. We aimed to find out the correlation of CTLA4 gene polymorphism and GO. A total of 42 participants were enrolled in this study, consisting of 22 patients with GO and 20 healthy controls. Chi-square or Fisher’s exact test were used to appraise the association between Graves’ ophthalmopathy and CTLA4 single nucleotide polymorphisms (SNPs). All regions of CTLA4 including promoter, exon and 3’UTR were investigated. There was no nucleotide substitution in exon 2 and exon 3 of CTLA4 region, and the allele frequencies of CTLA4 polymorphisms had no significant difference between patients with GO and controls. However, the genotype frequency of “TT” genotype in rs733618 significantly differed between patients with GO and healthy controls (OR = 0.421, 95%CI: 0.290–0.611, *p* = 0.043), and the “CC” and “CT” genotype in rs16840252 were nearly significantly differed in genotype frequency (*p* = 0.052). Haplotype analysis showed that CTLA4 Crs733618Crs16840252 might increase the risk of GO (OR = 2.375, 95%CI: 1.636–3.448, *p* = 0.043). In conclusion, CTLA4 Crs733618Crs16840252 was found to be a potential marker for GO, and these haplotypes would be ethnicity-specific. Clinical application of CTLA4 Crs733618Crs16840252 in predicting GO in GD patients may be beneficial.

## 1. Introduction

Graves’ disease (GD) is an autoimmune inflammatory disease. The annual incidence of Graves’ disease is 20 to 50 cases per 100,000 persons [1]. Graves’ ophthalmopathy (GO) also called thyroid eye disease occurs in 25–50% of patients suffered from GD [2]. The reported prevalence rates of Graves’ ophthalmopathy among different ethnic populations are limited. An early study demonstrated 6.4 times higher risk of developing GO in Caucasians than that of Asians [3] but has been criticized for a small sample size. Recent studies reported similar prevalence rate in Malaysians (Malay, Chinese, and Indian) [4] and Indians [5] with GD as compared with Caucasian GD patients. The clinical presentation of GO includes eyelid retraction, restrictive extraocular myopathy, proptosis, glaucoma, exposure keratitis and compressive optic neuropathy [6]. Patients may experience frequent diplopia, dry eye, chemosis, eyelid swelling and even vision loss in severe cases [7]. Additionally, patients with Graves’ disease tend to have concomitant hyperthyroidism.

Genetic analysis has identified several susceptible genes, including genes encoding thyroglobulin, thyrotropin receptor, HLA-DRβ-Arg74, protein tyrosine phosphatase nonreceptor type 22 (PTPN22), cytotoxic T-lymphocyte–associated antigen 4 (CTLA4), CD25, and CD40 [8]. Recent studies have identified circulating autoantibodies which target the thyroid-stimulating hormone receptor and insulin-like growth factor-1 receptor on the orbital fibroblasts [9,10]. Further activation of inflammatory genes leads to overproduction of cytokines, chemoattractants, hyaluronan and glycosaminoglycan, which results in orbital inflammation in the early phase and tissue hypertrophy in the late phase [11]. The expression level of these gene associated with GD and inflammation would different according to their genotype, and further study on genetic polymorphism in these genes would predict prognosis or complications in GD patients.

Treatment of GO varies according to the disease severity [12]. Establishing a euthyroid state and quitting smoking are helpful [13]. Active inflammatory phase often requires corticosteroids or other immunosuppressants to reduce orbital congestion [14]. Early intervention and close monitoring may hamper the progression and the severity of the disease [15]. It would be beneficial to find biomarkers early predicting severe GO in GD patients. Herein, all regions of CTLA4 including promoter, exon and 3′UTR were investigated to find the correlation of CTLA4 gene polymorphism and GO.

## 2. Experimental Section

### 2.1. Study Subjects

Twenty-two patients with GO (46.2 ± 16.5 years old, eight males and fourteen females) and 20 healthy controls (28.3 years old, 5 males and 15 females) were included in Linkou Chang Gung Memorial Hospital during the period from June 2017 to May 2019.

### 2.2. DNA Extraction

The blood samples were collected in EDTA-coated vacuum tubes. The genomic DNA was extracted using QIAamp^®^ DNA Mini kit (Qiagen GmbH, Hilden, Germany) according to the manufacturer’s instructions. DNA concentration and purity were evaluated by measuring the optical density at 260 and 280 nm through UV spectrometer.

### 2.3. PCR Amplification

The PCR mixture contained 1 µL DNA, 10µL HotStarTaq DNA Polymerase (Qiagen GmbH, Hilden, Germany), 1 µL forward primer (10 µM), 1 µL reverse primer (10 µM) and 12 µL ddH_2_O. Table 1 displayed the primers used in the present study. The PCR program for promoter and exon 1 was 1 cycle 95 °C for 10 min, 35 cycles of 94 °C for 30 sec, 65.5 °C for 30 sec, and 72 °C for 1 min. The PCR program for exon 2, exon 3, exon 4, and 3′UTR was 1 cycle 95 °C for 10 min, 35 cycles of 94 °C for 30 s, 59 °C for 30 s, and 72 °C for 1 min. The final elongation step was 3 min at 72 °C and then soaking at 10 °C. For gel electrophoresis visualization, 5 μL of the PCR products was pipetted onto a 1.5% agarose gel and run at 100 V for 20 min. Correct PCR products were visualized under UV illumination.

### 2.4. Purifying and Sequencing

The PCR production was purified by mixture containing 2.5 µL shrimp alkaline phosphatase and 0.05 µL exonuclease I (New England Biolabs, UK), and the purified PCR products were sequenced using ABI PRISM 3730 DNA Analyzer (Applied Biosystems, Foster City, CA, USA). Single nucleotide polymorphisms (SNPs) analysis were performed on CTLA4 gene, and rs11571315, rs733618, rs4553808, rs11571316, rs62182595, rs16840252, rs5742909, rs231775, and rs3087243 were selected for genotyping.

### 2.5. Statistical Analysis

All statistical data analyses were performed by the SPSS 17.0. Genotype and allele frequencies of CTLA4 gene for patients with GO were compared to healthy controls. Chi-square or Fisher’s exact test were used to evaluate the association between GO and CTLA4 SNPs.

## 3. Results

Among 22 GO patients, there were 14 females and 8 males, and the average age of patients was 46 years old. Moreover, half of GO patients was moderate in severity categories [16], and the most common symptom of GO patients was proptosis. The characteristics of patients were shown in Table 2.

The severity of GO was classified via the criterion proposed by Werner [6]. Class 0 to class 2 were classified as “mild”, class 3 and class 4 were classified as “moderate”, class 5 and class 6 were classified as “severe”.

All SNPs were in accordance with the Hardy-Weinberg equilibrium in the control group (*p* > 0.05) (Table 3). There was no nucleotide substitution in exon 2 and exon 3 of CTLA4 region, and the allele frequencies of CTLA4 polymorphisms had no significant difference between patients with GO and controls. However, the genotype frequency (Table 4) of “TT” genotype in rs733618 significantly differed between patients with GO and healthy controls (OR = 0.421, 95%CI: 0.290–0.611, *p* = 0.043), and the “CC” and “CT” genotype in rs16840252 were nearly significantly differed in genotype frequency (*p* = 0.052). Haplotype analysis (Table 5) showed that CTLA4 Crs733618Crs16840252 might increase the risk of GO (OR = 2.375, 95%CI: 1.636-3.448, *p* = 0.043).

Our data suggested that the genotype of −1722 (rs733618) in the promoter region of CTLA4 was associated with GO, the p value of TT vs. (CT + CC) was 0.043. We further combined −1722 (rs733618) and −1147 (rs16840252) together to observe the correlation between GO and controls, because the genotype of −1147 was very close to the critical point, *p* = 0.052. As a result, the two-marker combination was found to be more significantly related to GO (*p* = 0.025).

## 4. Discussion

Previous research had found that rs733618 [17], rs231775 [17], rs3087243 [18,19] and a two-marker combination containing rs733618 and rs16840252 [17] were related to GD. Our results showed that rs733618 was specifically related to GO. Comparing family-based research [17] to our results, only rs733618 in promoter was related to GO but not rs231775 in exon 1, and it indicated that major transcription level of CTLA4 was affected by SNPs in promoter region. Moreover, SNPs in exon 1 did not lead to significant reduced protein expression [20]. Therefore, rs733618 were associated with GD and GO, and “TT” genotype of rs733618 which was found to increasing the risk of GO in our study may affect major transcription level of CTLA4.

W. Tang et al., used expectation-maximization (EM) algorithm to create several feasible haplotypes, and haplotype compounded of Grs733618Crs16840252Ars231775Grs3087243 was associated with gastric cardia adenocarcinoma (*p* = 0.012) [21]. Our results indicated that Crs733618Crs16840252 was related to GO. Therefore, these two SNPs might be involved in both cancer and autoimmune disease. The immune dysfunction would be affected by CTLA4 genetic variation, and the risk of cancer and the development of autoimmune disease would be increased [22]. The rs16840252 mutation would increase the risk of cancer as well [23]. The most central function of CTLA4 is negative regulation of T cell activation, and it is expressed on the surface of activated T cells and competes to B7 with CD28 [24,25]. On the contrary, autoimmune disease would develop by reducing CTLA4 transcription level because of the immune responses are overreacting [26]. Investigation of rs733618 and rs16840252 in T cell activation would uncover the underlying mechanism in development of GO. In Table 6, disease association of rs733618 [23,27,28,29,30,31,32,33,34] and rs16840252 [35,36] were summarized.

Several studies in GD-related CTLA4 SNPs were summarized in Table 7. We found that those GD-related SNPs were located at different positions in the CTLA4 region within different populations. The majority of patients with GD from different ethnicities were associated with rs231775 and/or rs3087243 (Polish [37,38], Chinese [39,40,41,42,43,44], Saudi Arabian [45,46], Iranian [47,48], Italian [49], Japanese [50], and Taiwanese [17,18,19]). However, rs733618 and its relationship with GO have been found merely in Taiwan population. Thus, the correlation between GO and SNPs would diversely result from racial specificity, severity of GO and various inclusion criteria. Further investigation on racial specific GD-related SNPs associated with GO would be indicated.

Herein, Crs733618Crs16840252 of the CTLA4 promoter was shown to be associated with increasing risk of GO. Promoter polymorphism can influence gene transcription, as well as the disease severity [53]. The signal peptide is encoded by exon 1, and mutations in this exon suppress CTLA-4 protein expression [54]. Regulation of mRNA-based processes, such as mRNA localization, mRNA stability, translation, as well as regulation of protein features not encoded in the amino acid sequence are most commonly regulated by 3′UTR [55]. Consequently, there would be other significant SNPs located including but not limited to promoter region associated with GO. Further clinical application for prediction of GO in GD patients needs a feasibility study, and investigation of the mechanism of CTLA4 SNPs and haplotype of GO should be performed.

## 5. Conclusions

In conclusion, CTLA4 Crs733618Crs16840252 was found to be a potential marker for GO, and these haplotypes would be ethnicity-specific. Clinical application of CTLA4 Crs733618Crs16840252 in predicting GO in GD patients may be beneficial.

## Figures and Tables

**Table 1 jcm-08-01842-t001:** Primers used for CTLA4 in this study.

Primer	GC Content	Tm (°C)	Base Pair
promotor and exon1			
F: 5′ GGC AAC AGA GAC CCC ACC GTT 3′ R: 5′ GAG GAC CTT CCT TAA ATC TGG AGA G 3′	21/13 (62%) 25/12 (48%)	65.3 65.8	1234
F: 5′ CTC TCC AGA TTT AAG GAA GGT CCT C 3′ R: 5′ GGA ATA CAG AGC CAG CCA AGC C 3′	25/12 (48%)	65.8	1170
	22/13 (59%)	65.8	
Exon2 and exon3			
F: 5′ CAT GAG TTC ACT GAG TTC CC 3′ R: 5′ TAC CAC TGT CCT TCC TCT TC 3′	20/10 (50%)	58.4 °C	1034
	20/10 (50%)	58.4 °C	
Exon4			
F: 5′ CTA GGG ACC CAA TAT GTG TTG 3′ R: 5′ AGA AAC ATC CCA GCT CTG TC 3	21/10 (48%)	59.5	360
	20/10 (50%)	58.4	
3′UTR			
F1: 5′ CAG CTA GGG ACC CAA TAT GTG TTG AG 3′ R1: 5′ GTC AAG TCA ACT CAG ATA CCA CCA GC 3′ F2: 5′ GCT TGG AAA CTG GAT GAG GTC ATA GC 3′ R2: 5′ AGA GGA AGA GAC ACA GAC AGA GTT GC 3′	26/13 (50%)	59.5	1088
	26/13 (50%)	59.5	
	26/13 (50%)	59.5	1255
	26/13 (50%)	59.5	

F: forward primer; R: reverse primer.

**Table 2 jcm-08-01842-t002:** Characteristics of patients (*n* = 22).

	Total, No. (%)
Median age of the patients	46.2 ± 16.5
Sex of the patients	
Male	8 (36.4)
Female	14 (63.6)
Graves’ ophthalmopathy	
Mild	7 (31.8)
Moderate	14 (63.6)
Severe	1 (4.6)

**Table 3 jcm-08-01842-t003:** Allele frequencies in patients and controls and odds ratio of developing Graves’ ophthalmopathy.

SNP	Position	Allele	Minor Allele Frequency		HWE *p* Value	Odds Ratio	χ^2^ *p* Value
			Patient	Control		(95% CI)	
rs11571315	203866178	C/T	0.181	0.275	0.237	1.707 (0.607–4.799)	0.308
rs733618	203866221	T/C	0.409	0.500	0.670	1.444 (0.609–3.425)	0.403
rs4553808	203866282	A/G	0.068	0.175	0.638	2.899 (0.695–12.091)	0.182
rs11571316	203866366	A/G	0.114	0.100	0.884	1.154 (0.287–4.635)	1
rs62182595	203866465	A/G	0.068	0.175	0.638	2.899 (0.695–12.091)	0.182
rs16840252	203866796	C/T	0.068	0.200	0.535	3.417 (0.838–13.927)	0.074
rs5742909	203867624	C/T	0.068	0.175	0.638	2.899 (0.695–12.091)	0.182
rs231775	203867991	A/G	0.182	0.275	0.237	1.707 (0.607–4.799)	0.308
rs3087243	203874196	A/G	0.114	0.100	0.884	1.154 (0.287–4.635)	1

HWE: Hardy-Weinberg equilibrium; 95%CI: 95% confidence interval.

**Table 4 jcm-08-01842-t004:** Statistical analysis of CTLA4 (SNPs).

SNP	Genotype	Genotype Frequency		Odds Ratio (95 % CI)	*p* Value
		Patient (*n*)	Control (*n*)		
rs11571315	CC	0	0	NA	NA
	CT	8	11	0.468 (0.136–1.611)	*p* = 0.226
	TT	14	9	2.139 (0.621–7.370)	*p* = 0.226
rs733618	CC	4	4	0.889 (0.190–4.150)	*p* = 1
	CT	18	12	3.000 (0.736–12.227)	*p* = 0.118
	TT	0	4	0.421 (0.290–0.611)	*p* = 0.043 ******
rs4553808	AA	19	13	3.410 (0.742–15.677)	*p* = 0.152
	AG	3	7	0.293 (0.064–1.348)	*p* = 0.152
	GG	0	0	NA	NA
rs11571316	AA	0	0	NA	NA
	AG	5	4	1.176 (0.267–5.176)	*p* = 1
	GG	17	16	0.850 (0.193–3.739)	*p* =1
rs62182595	AA	0	0	NA	NA
	AG	3	7	0.293 (0.064–1.348)	*p* = 0.152
	GG	19	13	3.410 (0.742–15.677)	*p* = 0.152
rs16840252	CC	19	12	1.22 (0.932–19.131)	*p* = 0.052 *****
	CT	3	8	0.237 (0.052–1.073)	*p* = 0.052 *****
	TT	0	0	NA	NA
rs5742909	CC	19	13	3.410 (0.742–15.677)	*p* = 0.152
	CT	3	7	0.293 (0.064–1.348)	*p* = 0.152
	TT	0	0	NA	NA
rs231775	AA	0	0	NA	NA
	AG	8	11	0.468 (0.136–1.611)	*p* = 0.226
	GG	14	9	2.139 (0.621–7.370)	*p* = 0.226
rs3087243	AA	0	0	NA	NA
	AG	5	4	1.250 (0.283–5.525)	*p* = 1
	GG	17	16	0.800 (0.181–3.536)	*p* = 1

95%CI: 95% confidence interval; NA: Not applicable. * indicates *p* < 0.1, ** indicates *p* < 0.05.

**Table 5 jcm-08-01842-t005:** *CTLA4* haplotypes and odds ratio of developing Graves’ ophthalmopathy.

CTLA4 Haplotypes	Patient(n)	Control (n)	OR (95%CI)	*p* Value
C_rs733618_C_rs16840252_	22	16	2.375 (1.636–3.448)	0.043 ******
C_rs733618_T_rs16840252_	3	5	0.474 (0.097–2.307)	0.445
T_rs733618_C_rs16840252_	18	16	1.125 (0.241–5.252)	1
T_rs733618_T_rs16840252_	3	7	0.293 (0.064–1.348)	0.152

OR: odds ratio; 95%CI: 95% confidence interval; NA: Not applicable. ** indicates *p* < 0.05.

**Table 6 jcm-08-01842-t006:** Summary of rs733618 and rs16840252-related disease.

SNP	Disease	Subjects and Results	Ref.
rs733618	Systemic lupus	Asian:	[23]
	erythematosus	C allele was strongly associated with SLE and also CC genotype was significantly associated with the risk of SLE, *p* = 0.000).	
	Breast cancer	Chinese:	[27]
		CC genotype and C allele showed an increased risk of breast cancer (*p* = 0.030, odds ratio (OR) = 1.457, 95% confidence internal (CI) 1.036–2.051; *p* = 0.024, OR = 1.214, 95% CI 1.026–1.436, respectively).	
	Polycystic ovary syndrome	Chinese Han population:	[28]
		significantly different between case and control groups in either genotypic or allelic distribution, *p* = 0.01 and 0.009, respectively.	
	Survival in patients with sepsis	Adult Caucasian patients with sepsis:	[29]
		lower 90-day mortality was observed for T_rs733618_ A_rs231775_ A_rs3087243_ haplotype-negative patients than for patients carrying the TAA haplotype, *p* = 0.0265.	
	Survival in patients with multiple myeloma receiving bortezomib-based regimens	Unrelated Chinese Han population: GG genotype reduced the progression-free survival and the overall survival of patients with multiple myeloma who received bortezomib-based therapy, *p* = 0.002.	[30]
	Non-small-cell lung cancer	Chinese:	[31]
	(NSCLC)	T > C polymorphism was associated with the development of NSCLC in ≥60 years and even drinking subgroups.	
	Myasthenia gravis (MG)	Chinese Han population:	[32]
		C allele were more frequent in MG patients, *p* = 0.042.	
	Urinary schistosomiasis	Gabonese children:	[33]
		T allele and TT genotype were significantly overrepresented in the patient group, *p* = 0.001.	
	Lymphatic filariasis (LF)	Sarawak, Malaysia:	[34]
		CT genotype (*p* = 0.02) and those with combined minor allele C carriers (CT + CC; *p* = 0.01) exhibited a significantly decreased risk for LF.	
rs16840252	Colorectal cancer	Chinese:	[35]
		polymorphism was associated with an increased risk of colon cancer in homozygote model *p* = 0.040 and recessive model *p* = 0.037.	
	Recurrent schizophrenia	Chinese Han population:	[36]
		A significant association with schizophrenia, *p* (allele) = 0.0081, *p* (genotype) = 0.0117.	

**Table 7 jcm-08-01842-t007:** Summary of several studies for *CTLA4* SNPs associated with Graves’ disease.

SNP	Subjects	Results	Ref.
rs733617	Han population of Taiwan (family-based)	C allele over-transmitted to affected individuals (χ^2^ = 6.714, nominal *p* = 0.0096).	[17]
rs5742909	Polish Caucasian	genotype and allele were differentially distributed (*p* = 0.0002; *p* = 4.07 × 10^−5^), and lack of the rare T allele increased GD in patients with familial autoimmune thyroid incidence (*p* = 0.00005).	[37]
	Chinese	genotype frequencies of CT and allele frequencies of T were much higher in GD patients with ophthalmopathy than that in the group without ophthalmopathy (*p* = 0.020, *p* = 0.019).	[39]
	Chinese	variant allele carriers might have decreased risks of GD when compared with the homozygote carriers TT + TC vs. CC: OR = 0.78, 95% CI = 0.62–0.97.	[44]
rs231775	Han population of Taiwan (family-based)	CTLA4_+49_G/A (*p* = 0.0219), with its minor allele (G allele) over-transmitted to affected individuals (χ^2^ = 5.252, nominal *p* = 0.0219).	[17]
	Taiwanese	significant differences in the frequencies of the genotypes and alleles, *p* < 0.05.	[18]
	Polish populations	a significantly lower frequency of the AA genotype in the group of patients with clinically evident GO (*p* = 0.02, OR = 2.6).	[38]
	Chinese	genotype GG and allele frequencies of G in patients with Graves’ disease were significantly increased as compared with control group (*p* = 0.008, *p* = 0.007).	[39]
	Han population of Chinese (unrelated)	allele G was significantly associated with GD in adults (*p* < 0.001) and children (*p* = 0.002)).	[40]
	Chinese children < 16 years old (unrelated)	genotype GG (*p* = 0.005) and allele G (*p* = 0.03) were more prevalent in GD.	[41]
	Chinese children	genotype and allele frequencies of children with GD differed significantly from those of the controls (*p* = 0.0023 and *p* = 0.022, respectively).	[42]
	Saudi Arabian	G allele was more frequent in patients with GD than in the control group, *p* = 0.003.	[45]
	Iranian	G allele was significantly higher in patients with Graves’ disease than in the control group (27.1% vs. 15.1%, OR = 2.096, 95%CI = 1.350–3.253 and *p* < 0.01).	[47]
	Iranian	a significant increase of GG genotype and G allele was observed in patients (*p* = 0.012 and *p* = 0.025, respectively).	[48]
	Italian	G allele frequency was significantly higher compared to control subjects (*p* = 0.04).	[49]
	Caucasian and Asian	G allele vs. A allele, *p* < 0.00001; genotype: GG vs. AG + AA, *P* < 0.00001; GG + AG vs. AA, *p* < 0.00001; GG vs. AA, *p* < 0.00001; AG vs. AA, *p* < 0.00001.	[50]
rs3087243	Taiwanese children	genotype GG was significantly associated with GD (OR = 1.71, 95% CI 1.20–2.44, *p* = 0.006); Allele G was significantly more frequent (OR = 1.61, 95% CI 1.18–2.19, *p* = 0.0049).	[19]
	Taiwanese	G allele is associated with susceptibility to Graves’ disease (*p* = 0.011).	[18]
	Han population of Chinese (unrelated)	allele G was significantly associated with GD in adults (*p* < 0.001) and children (*p* < 0.001).	[40]
	Chinese children < 16 years old (unrelated)	G allele was more prevalent in GD *p* = 0.02.	[41]
	Southern China	G allele was significantly associated with an increased risk of GD development, *p* < 0.001.	[43]
	Chinese	G > A allele frequencies between the patient and control groups, *p* = 0.014.	[44]
	Russian	A allele and the AA genotypes were significantly increased in patients with GD.	[45]
	Saudi Arabian	G allele was higher in GD patients than those in controls, *p* = 0.004.	[46]
	Italian	allelic frequency of the G allele was also significantly higher in patients with GD (*p* = 0.02).	[49]
	Japanese	for the TBII-positive GD, G allele carriers in patients had significant association with GD, OR = 2.97, 95%CI = 1.29–6.87, *p* = 0.008.	[51]
	Russian	significantly higher frequencies of A allele and AA genotype and a lower proportion of G allele and GG genotype.	[52]

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
