# Peer review of "Investigation of the Correlation between Graves’ Ophthalmopathy and CTLA4 Gene Polymorphism"

_jcm, 2019, doi:10.3390/jcm8111842_

Round 1

Reviewer 1 Report

The introduction should have more details. Discussion is too short without any clear conclusions. I did not find clear statements.

This paper is very interesting and well conducted, but in my opinion this paper has low clinical relevance. Thus I suggest another journal focusing on experimental studies. 

Author Response

Point 1: The introduction should have more details. Discussion is too short without any clear conclusions. I did not find clear statements. This paper is very interesting and well conducted, but in my opinion this paper has low clinical relevance. Thus I suggest another journal focusing on experimental studies.

Response 1: We appreciated the reviewer’s comment. We have added more details to introduction and discussion. Clear statements were made in conclusions.

Reviewer 2 Report

A very well written paper.  Very interesting!

1-Please reformat the references to be more consistent.

2-Table 2 - need to add the characteristics of the controls.

3-Line 93 - Werner (reference 13), not reference 6.

4-Paragraph 2 of Discussion, reference 21 should be listed before reference 22 (need to re-number).

5-Are there any associations with increased risk of thyroid nodules or thyroid cancers in patients with GD or GO.

6-Table 7:  delete d.=1 (3rd paragraph).

Author Response

A very well written paper. Very interesting!

Author reply: We appreciated the reviewer’s comment.

Point 1: Please reformat the references to be more consistent.

Response 1: We have revised them.

Point 2: Table 2 - need to add the characteristics of the controls.

Response 2: We have added and revised them in the study subjects section of experimental section.

Point 3: Line 93 - Werner (reference 13), not reference 6.

Response 3: We have revised them.

Point 4: Paragraph 2 of Discussion, reference 21 should be listed before reference 22 (need to re-number).

Response 4: We have revised them.

Point 5: Are there any associations with increased risk of thyroid nodules or thyroid cancers in patients with GD or GO.

Response 5: Studies have suggested an increased risk of thyroid malignancy in Graves' disease. (Staniforth JU, Erdirimanne S, Eslick GD. Thyroid carcinoma in Graves' disease: A meta-analysis. Int J Surg. 2016 Mar;27:118-25.)

Point 6: Table 7: delete d.=1 (3rd paragraph).

Response 6: We have revised them.
